# The alarmin IL33 orchestrates type 2 immune-mediated control of thymus regeneration

Emilie J. Cosway[1], Kieran D. James [1], Andrea J. White[1], Sonia M. Parnell [1], Andrea Bacon[1], Andrew N. J. McKenzie[2], W. E. Jenkinson[1] & Graham Anderson [1] ✉

As the primary site of T-cell development, the thymus dictates immune competency of the host. The rates of thymus function are not constant, and thymus regeneration is essential to restore new T-cell production following tissue damage from environmental factors and therapeutic interventions. Here, we show the alarmin interleukin (IL) 33 is a product of Sca1[+] thymic mesenchyme both necessary and sufficient for thymus regeneration via a type 2 innate immune network. IL33 stimulates expansion of IL5-producing type 2 innate lymphoid cells (ILC2), which triggers a cellular switch in the intrathymic availability of IL4. This enables eosinophil production of IL4 to re-establish thymic mesenchyme prior to recovery of thymopoiesis-inducing epithelial compartments. Collectively, we identify a positive feedback mechanism of type 2 innate immunity that regulates the recovery of thymus function following tissue injury.

The production and selection of αβT-cells in the thymus is an essential feature of the vertebrate adaptive immune system. During this process, intrathymic T-cell development is initiated by thymus colonisation of lymphoid progenitors and culminates in the generation of self-tolerant major histocompatibility complex (MHC)-restricted CD4[+] and CD8[+] αβT-cells that emigrate from the thymus and populate peripheral tissues[1–3]. Thus, thymus function enables the host to mount effective immunity against pathogens and cancers, and benefit from therapeutic interventions including successful vaccination strategies. Importantly, the process of intrathymic T-cell production is not cell-autonomous and instead depends upon complex interactions between developing thymocytes and multiple thymic stromal compartments. These include a diverse range of thymic epithelial cells (TEC) as well as mesenchyme and endothelium that collectively control both thymocyte development and tolerance induction[4,5]. Importantly, although the thymus remains functional throughout life, the rates of thymopoietic function and T-cell production are variable. For example, while chronic age-related thymic involution results in a progressive decline in T-cell development, the thymus is also highly sensitive to stimuli

that cause acute tissue damage, resulting in detrimental effects on thymus function[6–8]. Perhaps most significantly, chemo- and radiotherapeutic interventions used in cancer treatment provoke the loss of thymic stromal environments, which disrupts thymus function and causes a decline in T-cell production and increased patient morbidity and mortality due to secondary immunodeficiency[9]. To combat this, the thymus possesses effective endogenous regeneration mechanisms that restore T-cell production. Here, the recovery of thymus function occurs via the effective restoration of the intrathymic stromal microenvironments that are required to re-establish new waves of thymopoiesis. In our previous studies, we identified an essential role for eosinophils in thymus regeneration[10], findings which highlighted the importance and potential of targeting the innate immune system to therapeutically enhance thymus recovery. Here, we have examined the cellular and molecular mechanisms that control eosinophil-dependent thymus regeneration, and studied the effector role of eosinophils in this process. We show that the alarmin interleukin (IL) 33 is expressed by a Sca1[+] subset of thymic mesenchyme cells and is essential for thymus regeneration. IL33 influences the recovery of thymus function

[1]Institute of Immunology and Immunotherapy, University of Birmingham, Birmingham, UK. [2]MRC Laboratory of Molecular Biology, Cambridge, UK. ✉e-mail: g.anderson@bham.ac.uk

during regeneration through the targeted expansion of intrathymic type 2 innate lymphoid cells (ILC2), known producers of the cytokine IL5. Increased availability of IL5[+] ILC then promotes an expansion of thymic eosinophils, which shunts intrathymic production of the type 2 cytokine IL4 away from CD1d-restricted natural killer (NKT)-cells and towards eosinophils. Finally, we show that IL4 is required for recovery of IL33[+] thymic mesenchyme following damage, indicating that IL4 is the effector molecule of eosinophils in thymus regeneration, which operates by restoring IL33[+] mesenchyme as a positive feedback loop. Collectively, identification of this intrathymic innate network that restores thymus function following injury provides new insights into how this primary lymphoid organ is controlled by innate immune components, and offers direction towards new therapies for immune recovery following tissue damage.

## Results

### The alarmin IL33, but not IL25, drives thymus regeneration by targeting ILC2

In multiple tissues, the alarmin cytokine family are potent regulators of an innate immune network that involves ILC2 and eosinophils[11–14]. Given the importance of both these cell types in thymus regeneration[10], we examined the importance of the alarmins IL25 and IL33 in the context of a sublethal irradiation model (SLI) of thymus regeneration[7,10]. Here, *Il25[−/−]* and *Il33[−/−]* mice were treated with SLI alongside WT controls, and harvested at d35 post-SLI, a point that represents the timing of full thymus recovery[10]. In these experiments (Fig. 1), *Il25[−/−]*, *Il33[−/−]* and WT control Balb/c mice were born and maintained within our animal facility to ensure correct comparison between experiments involving WT and cytokine deficient mice. Interestingly, while analysis of TEC (identified as in S. Fig. 1) and T-cell development (S. Fig. 2) in mice lacking IL25 showed no alterations in

thymus recovery compared to WT controls (Fig. 1a), mice lacking IL33 showed impaired thymus regeneration, with diminished numbers of TEC and thymocytes (Fig. 1b). Thus, IL33, but not IL25, is essential for thymus regeneration after SLI treatment. To further examine the relative importance of alarmins in thymus recovery we compared the ability of in vivo administration of recombinant IL25 and IL33 to boost thymus regeneration after SLI (Figs. 2a, 3a). In these experiments (Figs. 2, 3), because of the need to obtain large cohorts of mice for experimental use, we used commercially available WT Balb/c mice that were housed in our animal facility following purchase, for the duration of the experiment to allow for comparison of mice injected with either PBS, IL25 or IL33. Here, we saw significant improvement of TEC and thymocyte recovery at d7 and d35 post-SLI in IL33-treated (Fig. 2b, c), but not IL25-treated mice (Fig. 3b, c). Moreover, confocal analysis of thymic tissue sections showed that enhanced thymus regeneration promoted by IL33 treatment occurred in the context of organised cortical and medullary thymic microenvironments (S. Fig. 3). Collectively, through use of both gain of function (cytokine injection) and loss of function (gene knockout mice), where experimental and control mice were suitably matched in terms of source and animal husbandry, our findings demonstrate a selective requirement for the alarmin IL33 in thymus regeneration and indicate IL33 therapy may provide opportunities to enhance the recovery of thymus function.

We next aimed to determine the mode of action of IL33 by examining effects on a series of genetically manipulated mice. Given that mice used in the experiments above are of a Balb/c background, and as additional mice of interest were readily available on a C57BL/6 background, we first needed to compare the ability of IL33 to boost thymus regeneration in both WT C57BL/6 and Balb/c mice. Importantly, administration of IL33 following SLI treatment resulted in

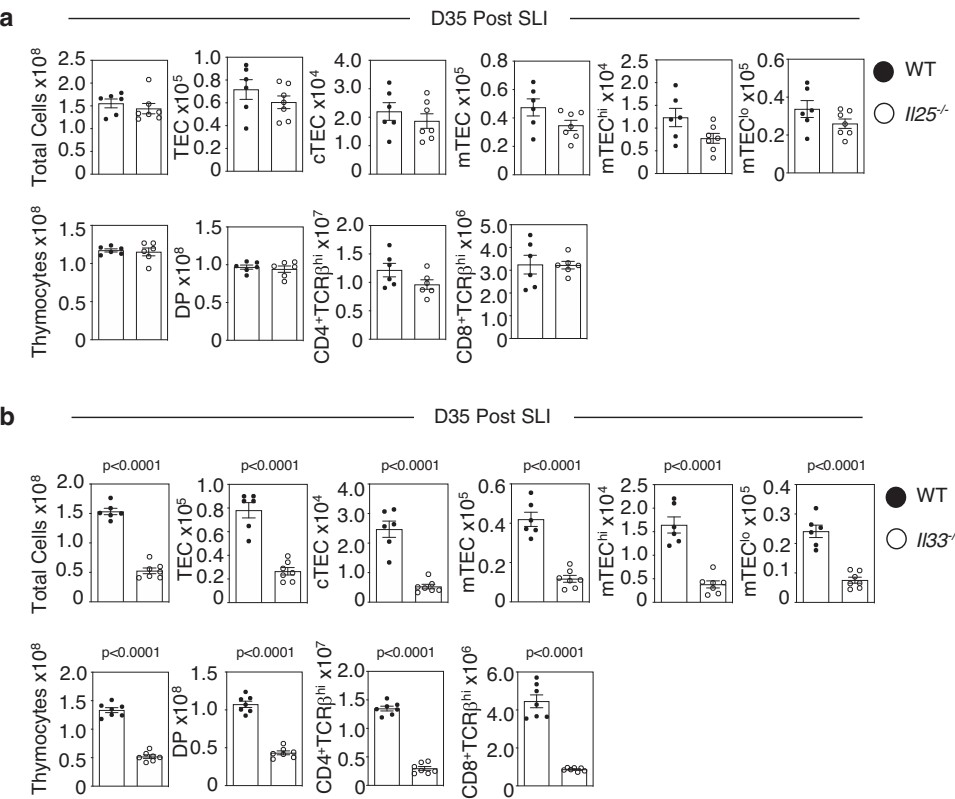

**Fig. 1 | Selective requirement for the alarmin IL33 in thymus regeneration.** Analysis of thymus regeneration (indicated TEC and thymocyte subsets) at 35 days post-SLI in WT (black circle) and either *Il25[−/−]* (**a**) or *Il33[−/−]* (**b**) mice. For (**a**), TEC WT *n* = 6 and *Il25[−/−]* *n* = 7 and thymocytes WT and *Il25[−/−]* *n* = 6 animals over 2

independent experiments. For (**b**), TEC WT *n* = 6 and *Il25[−/−]* *n* = 7 and thymocytes WT and *Il25[−/−]* *n* = 7 animals over 2 independent experiments. All error bars show the mean ± SEM. *P*-values were obtained using two-tailed, unpaired Student's *t* tests.

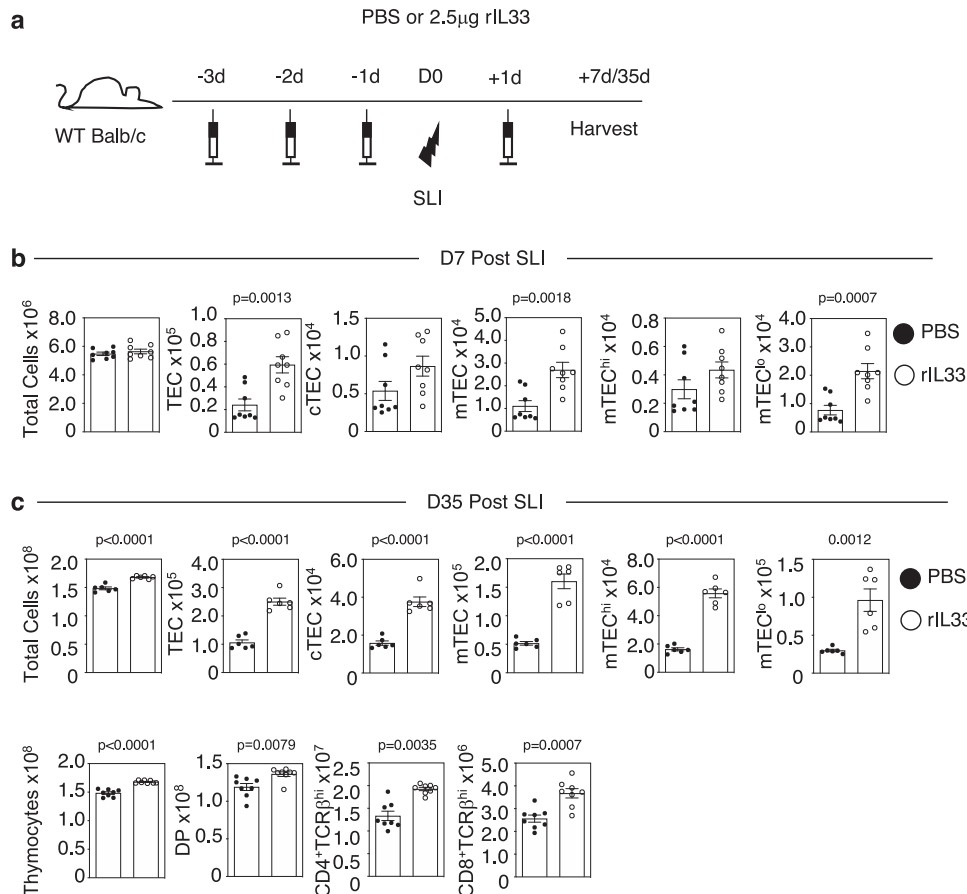

**Fig. 2 | IL33 boosts thymus regeneration. a** Injection regime for the administration of recombinant IL33 (rIL33), 2.5μg/mouse/injection or PBS control. Three injections were given prior to sub-lethal irradiation (SLI) (1 × 425 rad) and one injection post-SLI exposure. Mice were harvested either 7 or 35 days after SLI. **b** Effects of IL33 administration on TEC recovery at 7 days post-SLI, and TEC and thymocyte recovery at 35 days post-SLI (**c**). For (**b**), PBS and rIL33 $n = 8$ animals obtained over 2 independent experiments. For (**c**), TEC PBS and rIL33 $n = 6$ and thymocytes PBS and rIL33 $n = 8$ animals obtained over 2 independent experiments. All bars show mean ± SEM. *P*-values were obtained using two-tailed, unpaired Student's *t* tests.

enhanced thymus regeneration in both C57BL/6 and Balb/c mice (S. Fig. 4), demonstrating that IL33 is a regulator of thymus regeneration in both strains. From this, we next investigated the impact of IL33 administration on ILC2 (identified as in S. Fig. 5), known targets of IL33 in the context of type 2 immune responses. We used C57BL/6 Red5/Rag2GFP[10,15,16] mice (Fig. 4a) to enable the detection of Lin⁻GFP⁻Red5⁺IL7Rα⁺KLRG1⁺NK1.1⁻ intrathymic ILC2 (Fig. 4b). Strikingly, we saw that while overall thymus cellularity remained unchanged, IL33 treatment resulted in a massive expansion (43-fold) in ILC2 numbers, including IL5^Red5+ ILC2 (Fig. 4b, c). In contrast, and consistent with the importance of IL33 but not IL25 for thymus regeneration shown earlier, IL25 administration had far less impact, resulting in only a modest increase (1.4-fold) in ILC2 numbers (Fig. 4b, c). To directly examine the relationship between IL33, ILC2 and their roles in thymus regeneration, we performed SLI treatment of *Il7ra^Cre^Rora^fl/fl* ILC2 deficient mice[12] that also received recombinant IL33 (Fig. 5a). *Il7ra^Cre* mice were used as controls, and in line with the effects of IL33 in WT mice, we saw an increase in thymus regeneration at d35 post-SLI in *Il7ra^Cre* control mice (Fig. 5b). Importantly, IL33 treatment of ILC2 KO mice did not enhance thymus recovery (Fig. 5b). Collectively, these findings demonstrate that IL33 is a potent regulator of thymic ILC2, and the ability of IL33 to enhance thymus regeneration is ILC2-dependent. Given this ability of IL33 to expand ILC2, and the production of IL5 by these cells, we examined effects of IL33 administration on eosinophils (identified in S. Fig. 6) in both *Il7ra^Cre* control and ILC2 KO mice following SLI. Flow cytometric analysis showed that while IL33 treatment resulted in a large increase (38-fold) in eosinophils numbers in *Il7ra^Cre* control mice

at d7 post-SLI, this was significantly diminished in ILC2 KO mice (Fig. 5c). Thus, IL33-mediated expansion of IL5-producing ILC2 promotes an increase in intrathymic eosinophils, known regulators of thymus generation following damage.

## Intrathymic Expression Of IL33 Is Restricted To Sca1⁺ Mesenchyme

In multiple tissues including gut and lung, expression of the alarmin IL33 maps to stromal cells, including epithelial cells, mesenchyme, and endothelium[14,17]. To identify the cellular source of IL33 in thymus, we used IL33^cit reporter mice[17] in association with a panel of markers that identify distinct thymic cell types. Consistent with other tissues, we found IL33^cit expression in the thymus was detectable only within CD45⁻ stromal, and not CD45⁺ haemopoietic cells (Fig. 6a). Analysis of specific thymic stromal elements (S. Figs. 1 and 7) identified compartmentalisation of expression, with IL33^cit detectable within CD45⁻EpCAM1⁻CD3I⁻ mesenchyme but not EpCAM1⁺ TEC, CD3I⁺ endothelium or integrin β7⁺ pericytes (Fig. 6b). Interestingly, in white adipose tissue, mesenchyme expression of IL33 has been shown to map to a Sca1⁺ subset of cells that are involved in tissue repair following injury[14,18]. Indeed, we found Sca1 was expressed by some but not all thymic mesenchyme, with IL33^cit exclusively expressed by the majority of Sca1⁺ cells (Fig. 6c). Thus, intrathymic expression of IL33 maps to a subset of thymic mesenchyme cells defined by Sca1 expression.

The thymus is an epithelial-mesenchymal organ, and during organogenesis in the embryo, mesenchymal cells are important in

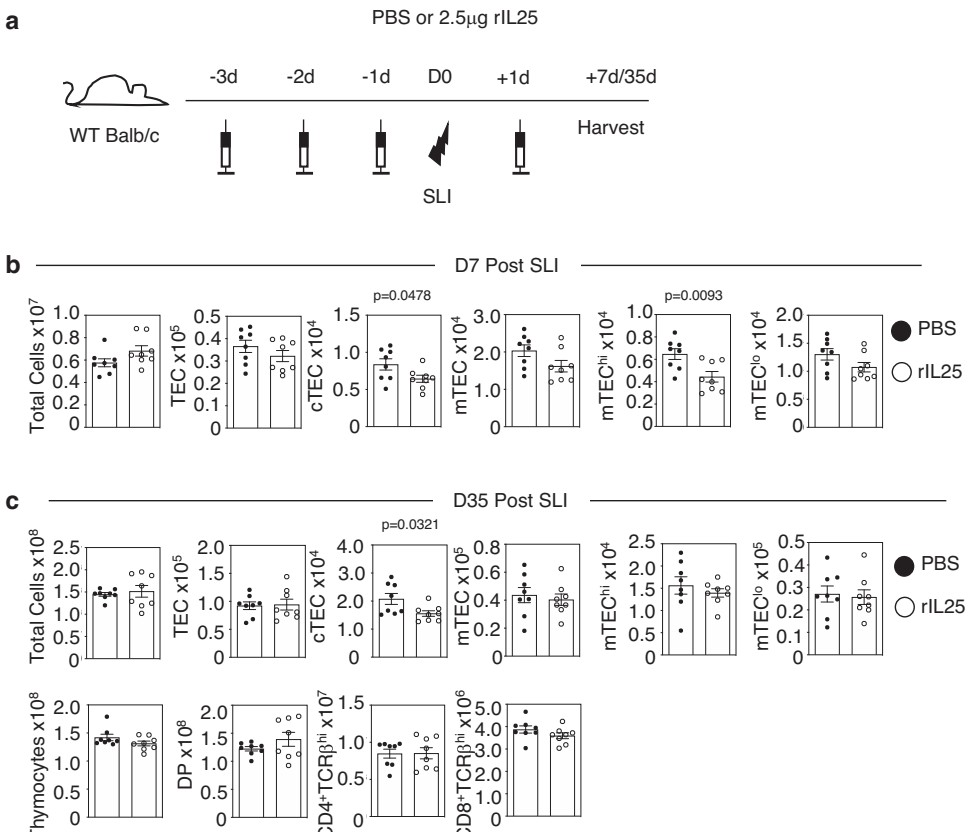

**Fig. 3 | IL25 does not enhance thymus regeneration. a** Injection regime for the administration of recombinant IL25 (rIL25), 2.5µg/mouse/injection or PBS control. Three injections were given prior to sub-lethal irradiation (SLI) (1 × 425 rad) and one injection post-SLI exposure. Mice were harvested either 7 or 35 days after SLI. **b** Effects of IL25 administration on TEC recovery at analysis 7 days post-SLI, and TEC

and thymocyte recovery at 35 days post-SLI (**c**). For (**b**), PBS and rIL25 $n = 8$ animals obtained over 2 independent experiments. For (**c**), TEC PBS and rIL25 $n = 8$ and thymocytes PBS and rIL25 $n = 8$ animals obtained over 2 independent experiments. All bars show mean ± SEM. $P$-values were obtained using two-tailed, unpaired Student's $t$ tests.

regulating the proliferation of thymic epithelial progenitors[19]. Thus, the establishment of functional TEC microenvironments that are required for T-cell development require thymic mesenchyme, which reinforces the long-known importance of epithelial-mesenchymal interactions in thymus development and function[20]. Given the expression of IL33 within thymic mesenchyme, we hypothesised that the mesenchymal compartment of the thymus may also be damaged during thymus injury. Consistent with this idea, and while haemopoietic cells are most sensitive to irradiation, several reports highlight that thymic stroma is also a target of irradiation[21–23]. To investigate this, and in particular mesenchymal cells following damage and during thymus regeneration, we analysed the kinetic of recovery of mesenchyme up to d35 post-SLI, where thymus cellularity is fully recovered (Fig. 6d). Interestingly, we saw a statistically significant decline in total thymic mesenchyme (S. Fig. 8a, b) and Sca1+ mesenchyme at d1 and d7 post-SLI, with numbers fully restored by d14 post-SLI (Fig. 6d, e). This demonstrates the sensitivity and regenerative capacity of Sca1+ mesenchyme following thymus injury. Despite this decrease in Sca1+ thymic mesenchyme, their levels of IL33cit were increased at d1 post-SLI (S. Fig. 9a). Moreover, the kinetic of Sca1+ mesenchyme recovery was significantly faster than the recovery of TEC. Thus, while Sca1+ mesenchyme was fully re-established by d14 post-SLI, TEC remained low at d14 post-SLI and reached full recovery at d35 post-SLI (Fig. 6e). These findings demonstrate that a stepwise recovery of stromal cells occurs during thymus regeneration, with mesenchyme recovery occurring prior to the re-establishment of epithelial compartments that drive T-cell development. To investigate mechanisms of mesenchyme recovery, we examined the importance

of the IL4/IL4Rα type 2 cytokine axis, a known regulator of mesenchymal stroma[14]. Consistent with this, Sca1+ thymic mesenchyme expressed IL4Rα (Fig. 6f). Moreover, Sca1+ mesenchyme numbers were reduced in *Il4ra*-/- mice at steady state (Fig. 6g), and IL4 treatment of IL33cit mice at steady state showed increased numbers of Sca1+ thymic mesenchyme cells (S. Fig. 9b, c). Perhaps most importantly, we found that the administration of IL4 to WT SLI-treated mice (Fig. 6h, i) improved the recovery of Sca1+ mesenchyme and increased their numbers at d7 and d14 compared to PBS-treated controls (Fig. 6i). Thus, the type 2 cytokine IL4 controls the damage-induced regeneration of IL33-producing Sca1+ mesenchyme that, through its selective expression of IL33, is an important regulator of thymus recovery following injury.

### Eosinophil production of IL4 controls thymus recovery from damage

Given this importance of IL4, we next examined its cellular source by analysing GFP+ cells in the thymus of Balb/c IL4GFP mice[24] at steady state and following SLI. Consistent with previous reports, the majority of GFP+ cells at steady state were CD1d-restricted NKT-cells (Fig. 7a)[10,25]. However, we also detected a small population of GFP+ cells in the thymus of Balb/c IL4GFP mice that lacked reactivity with CD1d-tetramer and were identified as Siglec-F+ eosinophils (Fig. 7a). At 1d post-SLI, we saw an increase in the proportion of thymic GFP+ cells (Fig. 7b, c), and this was accompanied by a dramatic shift in their cellular makeup, with the vast majority of GFP+ cells now being eosinophils (Fig. 7c, d). Thus, thymus damage in Balb/c mice causes a shunt in the cellular source of IL4 from NKT cells to eosinophils. Further analysis of 4Get mice showed

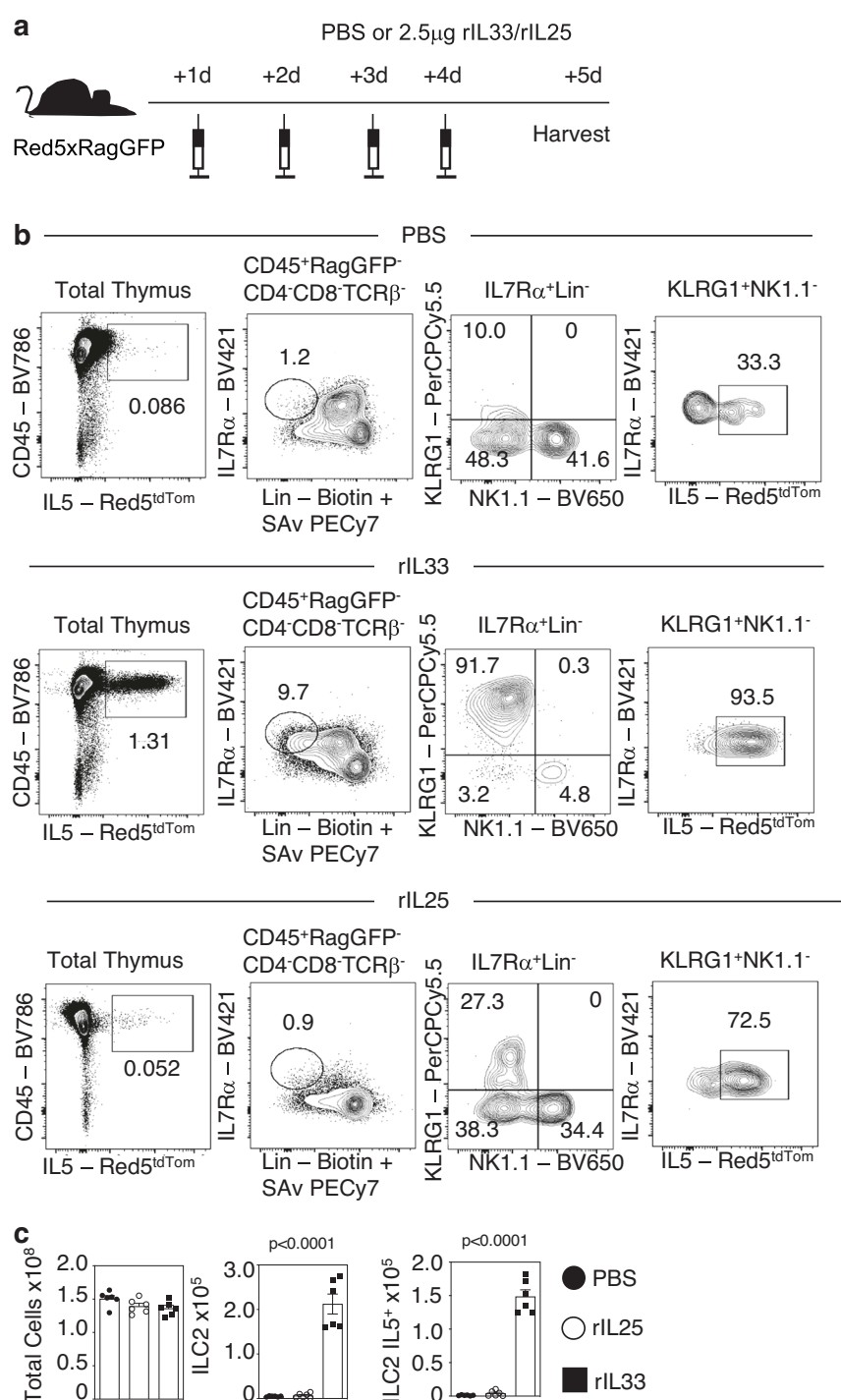

**Fig. 4 | IL33 Expands ILC2 In The Thymus. a** Injection regime for the administration of rIL33 or rIL25 in Red5xRagGFP mice, 2.5μg/mouse/injection or PBS control, four injections given. **b** Representative FACs plots, where *n* = 6, to illustrate ILC gating and effect of PBS/rIL33/rIL25 administration, total ILC are gated as CD45⁺RagGFP⁻CD4⁻CD8⁻TCRβ⁻IL7Rα⁺Lin⁻ cells, with ILC2 further identified as KLRG1⁺NK1.1⁻ cells. **c** Quantitation of effects of rIL33 and rIL25 administration on total thymus cellularity, ILC2 and IL5⁺ILC2 in Red5xRagGFP mice. Significance is of rIL33 or rIL25 treatment compared to PBS controls, (*n* = 6 animals for each condition). All analysis was carried out across two independent experiments. All statistical analysis was obtained using one-way ANOVA with Dunnett's multiple comparisons. All bars show mean ± SEM.

that IL4$^{GFP}$ expression was also detectable in thymic ILC2 (Fig. 7e), consistent with other reports describing ILC2 as a source of type 2 cytokines[26,27] in other tissues. Interestingly, ILC2 were undetectable in the thymus at 1-day post-SLI (Fig. 7e, f and S. Fig. 10), contrasting with eosinophil frequency which peaks at this timepoint[10]. The absence of thymic ILC2 at 1-day post-SLI suggests that their involvement in thymus regeneration is likely limited to an early (within 24 h) window after

damage induction, and suggests that despite their expression of IL4 in the steady state, ILC2 are not likely to operate as key providers of IL4 during the regeneration process. Given the evidence that eosinophils are a major source of intrathymic IL4 post-SLI, and the knowledge that Sca1⁺ mesenchyme cells are targets of IL4[14], we wondered whether Sca1⁺ mesenchyme was also affected in eosinophil deficient ΔdblGATA mice[28]. At steady state, numbers of Sca1⁺ mesenchyme were

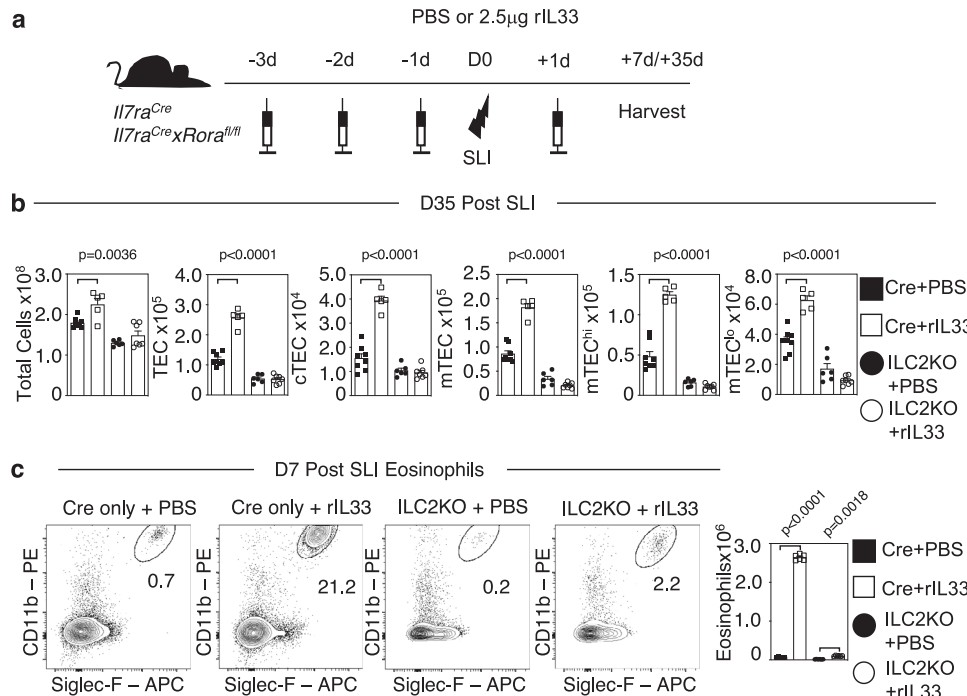

**Fig. 5 | IL33 expands ILC2 To enable intrathymic expansion of eosinophils.**
**a** Injection regime of *Il7ra^Cre* (control) and *Il7ra^Cre xRora^fl/fl* mice with PBS or 2.5μg rIL33, with three injections prior to SLI (1 × 500 rad) and then one injection post-SLI. **b** Quantitation of TEC recovery (*n* = 5–7) in control and ILC2 KO mice at 35 days post-SLI following PBS or rIL33 treatment. **c** Thymic eosinophil numbers in control and ILC2 KO mice at day 7 post-SLI and following PBS or rIL33 treatment (*n* = 6 animals from 2 independent experiments). Statistical significance is shown comparing *Il7ra^Cre*

(Cre) PBS (*n* = 8 animals from 4 independent experiments) with *Il7ra^Cre* (Cre) rIL33 (*n* = 5 animals from 4 independent experiments) or *Il7ra^Cre xRora^fl/fl*(ILC2KO) PBS (*n* = 6 animals from 4 independent experiments) with *Il7ra^Cre xRora^fl/fl*(ILC2KO) rIL33 (*n* = 7 animals from 3 independent experiments). Eosinophils were identified as CD45⁺CD4⁻CD8⁻TCRβ⁻TER119⁻CD11b⁺Siglec-F⁺. All statistical analysis was obtained using one-way ANOVA with Šídák's multiple comparisons. All bars show mean ± SEM.

comparable in ΔdblGATA and WT mice (Fig. 7g), which is consistent with the finding that iNKT cells, and not eosinophils, are the major source of IL4 in untreated mice (Fig. 7d). In contrast, while Sca1⁺ mesenchyme recovery was observed in the regenerating thymus of WT mice at 14d post-SLI, this did not occur in ΔdblGATA mice (Fig. 7g). This indicates that eosinophil-derived IL4 is essential for Sca1⁺ mesenchyme recovery after damage. Finally, to examine this directly, and understand the mode of action of eosinophils in thymus regeneration, we examined whether IL4 administration might replace the requirement for eosinophils in this process (Fig. 7h). We found that IL4 treatment of eosinophil deficient SLI-treated ΔdblGATA mice increased Sca1⁺ thymic mesenchyme numbers (Fig. 7i), and enhanced recovery of thymus size, TEC and thymocyte development (Fig. 7j, k). Importantly, IL4 treatment did not restore eosinophil development in ΔdblGATA mice (S. Fig. 11), arguing against the idea that this might explain their IL4-mediated improvement of thymus regeneration. Rather, these findings suggest that IL4 production by eosinophils acts directly on thymic stroma and provides an explanation for their importance in thymus regeneration.

## Discussion

The thymus plays an essential role in the development and function of the adaptive immune system by controlling αβT-cell production and selection. Central to thymus function are the epithelial microenvironments that support multiple stages of thymocyte development[3,29]. During thymus injury, TEC microenvironments are diminished, which results in impaired thymopoiesis and prevents effective recovery of the immune system. This is perhaps most significant during the treatment of haematological disorders, where thymus injury caused by pre-conditioning regimes results in defective immune reconstitution[3,30,31]. While detrimental effects of thymus

damage impact the immune competence of the host, endogenous thymus regeneration restores thymus function, resulting in the production of new naïve αβT-cells.

Here, we have the cellular and molecular mechanisms that control thymus regeneration following injury. In particular, we re-investigated the role played by eosinophils in thymus regeneration[10], which remains a poorly understood process. Here, we show that the alarmin IL33 plays a key role in the activation of an intrathymic type 2 immune network that culminates in the recovery of thymic stromal microenvironments via IL4 production by eosinophils. Specifically, IL33 expression by Sca1⁺ thymic mesenchyme promotes the expansion of IL5-producing ILC2, which then alters the intrathymic cellular sources of the type 2 cytokine IL4, which in turn regulates the recovery of IL4Rα expressing mesenchyme and ultimately thymic epithelial microenvironments. Important to this study and our previous work[10] is analysis of the intrathymic source of IL4. In our previous study using IL4^GFP reporter (4^Get) mice on a C57BL/6 background, we showed that at steady state, thymic GFP⁺ cells consisted mainly of NKT-cells. In our current study, we used the same 4^Get mice, but on a Balb/c background to ensure strain comparability with *Il4ra⁻/⁻*, ΔdblGATA mice used here. Interestingly, at steady-state in Balb/c 4^Get mice, as in C56BL/6 4^Get mice, the majority of GFP⁺ cells in thymus were NKT cells. However, following SLI irradiation of Balb/c 4^Get mice, the majority of GFP⁺ cells were eosinophils, which contrasts with a dominance of NKT cells in C57BL/6 mice after SLI[10]. Thus, SLI irradiation and thymus regeneration in Balb/c mice results in a shift in the cellular source in IL4 production towards eosinophils. This important requirement for the type 2 cytokine IL4 in Balb/c mouse models fits well with the known bias of this strain to type 2 immunity compared to C57BL/6 mice[32]. Furthermore, the finding that the intrathymic cellular sources of IL4 in Balb/c and C57BL/6 mice differ post-damage may indicate differences in the

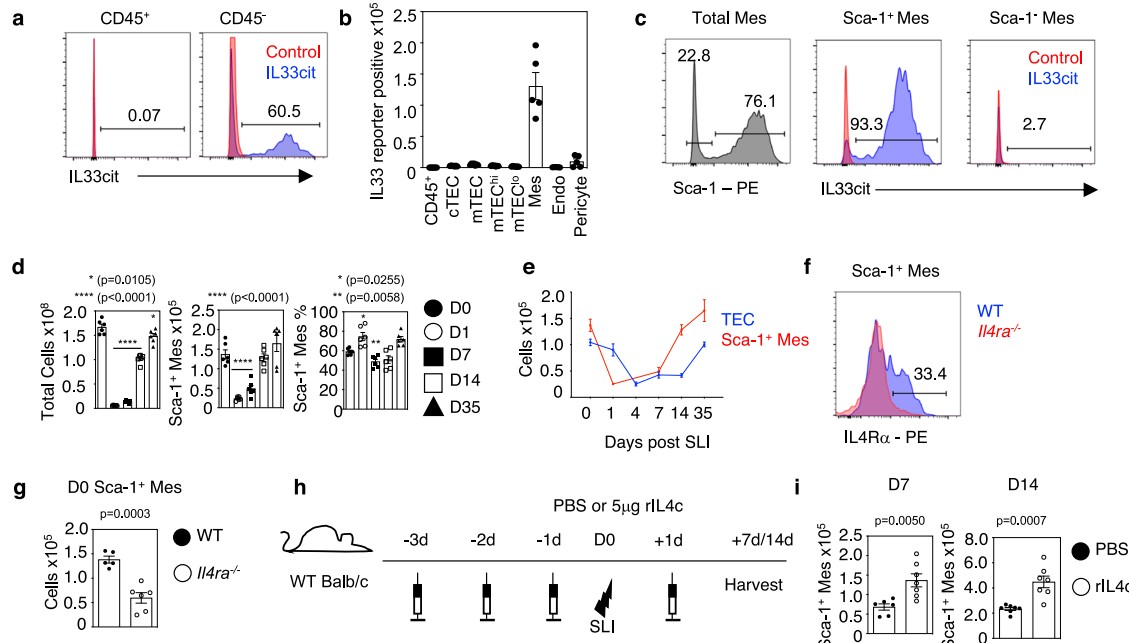

**Fig. 6 | IL33 expression selectively maps to Sca1± thymic mesenchyme.**
**a** Representative FACs plots illustrating the proportion of IL33cit expressing CD45+ thymic cells and total CD45- thymic stromal cells (blue histogram). Red histograms represent florescence levels in control WT mice. **b** Quantitation of IL33cit expression in defined thymic stromal populations; cTEC (CD45-EpCAM-1-UEA1-Ly51+), mTEC (CD45-EpCAM-1-UEA1+Ly51-), mTEChi (CD45-EpCAM-1-UEA1-Ly51-MHCII+CD80+), mTEClo(CD45-EpCAM-1-UEA1+Ly51-MHCII-CD80-), Endothelium (TER119-CD45-EpCAM-1-CD31+), Mesenchyme (TER119-CD45-EpCAM-1-CD31-Integrinα7-), Pericytes (TER119-CD45-EpCAM-1-CD31-Integrinα7+), (where n = 7 animals for TEC subsets and n = 5 animals for all other subsets, obtained from 2 independent experiments). **c** Representative FACs plots showing Sca-1 expression in thymic mesenchyme, and IL33cit expression within Sca-1+ and Sca-1- subsets. Blue histograms are from IL33cit mice, red histograms from control WT mice. **d** Total thymus cellularity, Sca-1+ mesenchyme numbers/proportions in WT mice following SLI exposure at indicated time points (n = 6 animals obtained from 2 independent experiments) with statistical analysis from a one-way ANOVA comparing means of

each group to D0 with Dunnett's multiple comparisons. **e** Comparative analysis of TEC (blue line) and Sca-1+ mesenchyme (red line) recovery at indicated time points in WT mice (n = 6 animals obtained from 2 independent experiments). **f** FACs plots showing IL4Rα expression in Sca-1+ mesenchyme from WT (blue histogram) mice, where against Il4ra-/- mice (red histogram) were used for control staining levels. **g** Quantitation of Sca1+ mesenchyme in WT and Il4ra-/- mice at steady state (D0) (WT n = 5 and Il4ra-/- n = 6 animals obtained from 2 independent experiments), from a two-tailed unpaired Student's t test. **h** Injection and irradiation regime of 5μg recombinant IL4-complexes/PBS into WT mice. Analysis of Sca-1+ mesenchyme was performed at day 7 and day 14 after injection regime, n = 6 minimum. **i** Numbers of Sca-1+ thymic mesenchyme cells at day 7 and day 14 post-SLI following PBS (D7 n = 6 and D14 n = 7 animals) or IL4c (D7 n = 7 and D14 n = 7 animals) injection, obtained from 2 independent experiments. Statistics from a two-tailed unpaired Student's t test. All analysis was carried out across at least two independent experiments. All error bars show mean ± SEM.

mechanisms of thymus regeneration in C57BL/6 and Balb/c mouse strains. This demonstrates the importance of mouse background in both the design of future studies and interpretation of previous studies, and indicates further work is required to examine how strain-specific differences occur during thymus regeneration.

Importantly, IL4 is necessary and sufficient to drive the recovery of Sca1+ mesenchyme following damage, creating a positive feedback loop for IL33-mediated control of tissue repair in the thymus. Importantly, we also show that IL4 administration can restore thymus regeneration in ΔdblGATA eosinophil deficient mice, which provides direct evidence that the requirement for eosinophils in thymus recovery is at least in part explained by their expression of this type 2 cytokine. Taken together, the ability of IL4 and IL33 to boost regeneration in Balb/c mice fits well with the idea that IL33 operates upstream in a mesenchyme>ILC2>eosinophil>mesenchyme pathway. Here mesenchyme production of IL33 expands thymic ILC2, which increases intrathymic IL5 availability that causes an increase in IL4+ eosinophils. Interestingly, analysis of ILC2 frequency after damage shows they are depleted from the thymus 1 day following SLI treatment, indicating that the requirement for, and involvement of, ILC2 in thymus regeneration occurs within a narrow time window. In contrast to the effects of IL33, IL4 likely operates downstream in thymus regeneration, with IL4 production from eosinophils forming part of a positive feedback loop that then aids regeneration of thymic mesenchyme. Interestingly, a direct comparison of the individual

effects of IL4 and IL33 showed that they exert similar positive effects on thymus regeneration. Moreover, a combination of IL4 and IL33 did not further enhance thymus recovery compared to either cytokine alone (S. Fig. 12). It is currently not clear whether IL4/IL33 administration exerts its effects solely by influencing events within the thymus, or whether it also involves targeting of cells in peripheral tissues that may also result in enhanced thymus recovery. However, the presence of targets in the thymus for IL33 (thymic ILC2) and IL4 (thymic mesenchyme) supports the possibility that positive effects of IL4/IL33 treatment can be explained at least in part by intrathymic mechanisms. Nevertheless, our current experiments do not rule out the impact of cytokine administration on extrathymic tissues. Thus, in addition to possible additional effects outside the thymus, we suggest that multiple stages of the intrathymic type 2 immune network can be triggered via cytokine administration at multiple points to equally boost thymus regeneration, perhaps offering a variety of opportunities for future therapeutic benefit.

Our finding that the importance of alarmins in thymus regeneration is limited to IL33 but not IL25 highlights the relevance of mesenchymal and not tuft cell products in this process. Here, it is important to note that our study focusses on elucidating mechanisms of thymus regeneration in the context of an acute model of thymus damage. Whether similar mechanisms are also important in other models where thymus function is diminished is not clear. For example, the thymus is also a known target of chronic injury, perhaps most

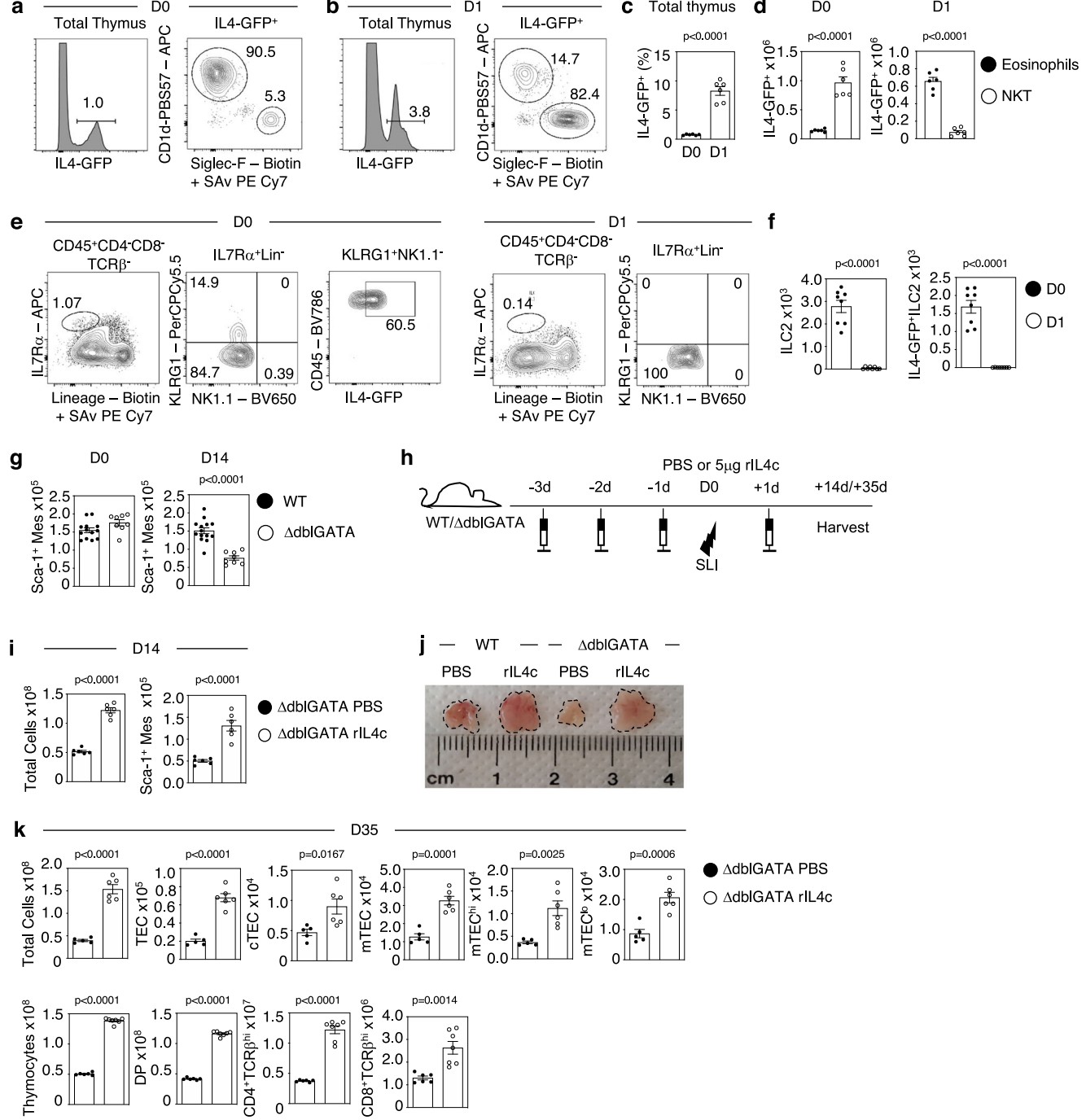

**Fig. 7 | Eosinophil production of IL4 controls thymus regeneration.**
**a** Representative FACs of total thymus cells from IL4^GFP mice, with analysis of CD1d-PBS57 tetramer and Siglec-F expression shown in GFP cells at D0 (**a**) and D1 post-SLI (**b**). **c, d** Show quantitation of total IL4^GFP, eosinophils and NKT cells within thymus at day 0 and day 1 post-SLI, $n = 6$ animals obtained from 2 independent experiments. **e** Representative FACs plots identifying ILC2 in thymus of IL4-GFP mice at D0 and D1 post-SLI. **f** Quantitation of ILC2 and IL4-GFP^+ ILC2 in thymus of IL4-GFP mice D0 (black circle) and D1 post SLI (white circle), $n = 8$ animals obtained from 2 independent experiments. **g** Quantitation of Sca-1^+ mesenchyme in WT (black circle) and ΔdblGATA (white circle) mice at day 0 (WT $n = 13$ from 4 independent experiments, ΔdblGATA $n = 8$ from 2 independent experiments) and day 14 (WT $n = 14$ from 4 independent experiments, ΔdblGATA $n = 8$ from 2 independent

experiments) post-SLI. **h** Injection regime of PBS or 5μg rIL4-complex into WT/ΔdblGATA mice, three injections prior to SLI and one after, mice harvested 14 days or 35 days post-SLI. **i** Analysis day 14 post-SLI of Sca-1^+ mesenchyme in ΔdblGATA injected as (**h**) ($n = 6$ animals obtained from 2 independent experiments). **j** Images show gross thymus size at day 35 post-SLI in WT and ΔdblGATA mice injected with either PBS or rIL4c. **k** Analysis at day 35 post-SLI of total thymus cellularity, and TEC and thymocyte subsets in ΔdblGATA mice following either PBS (black circle) or IL4c injection (white circle), where for TEC analysis PBS $n = 5$ and IL4c $n = 6$ and for thymocyte analysis PBS $n = 6$ and IL4c $n = 7$ obtained from 2 independent experiments. Statistical analysis was performed using a two-tailed unpaired Student's $t$ test. All significance shows mean ± SEM.

notably age-related thymus involution[33–35]. Further examination of the importance of type 2 immune networks described here in other models of long-term thymus damage warrants future study.

Previous studies have shown the importance of mesenchyme in regulating the growth of TEC microenvironments that support T-cell development[19,36]. While several studies have examined the effects of damage on thymus stroma, including irradiation-induced damage, most studies relate to the impact on TEC[21–23], with effects on non-TEC stroma being less clear. Here, we show that thymic mesenchyme is also a target for damage, which extends our understanding of the effects of irradiation-induced damage on thymus microenvironments. Furthermore, these findings indicate that changes in thymus function following injury are likely caused by a combinatorial effect on epithelial and mesenchyme compartments. Importantly, we show that thymic mesenchyme is also capable of regeneration. This recovery of thymic mesenchyme is rapid, and occurs prior to TEC recovery. Significantly, recovery of thymic mesenchyme can be boosted via IL4 treatment, a finding that represents a new approach to manipulate the mesenchymal component of the thymus. Given the step-wise recovery of mesenchyme and epithelial cells during thymus regeneration, our findings support a model in which the re-growth of TEC microenvironments after damage is determined by the earlier re-establishment of thymic mesenchyme, which then controls the recovery of cortical and medullary epithelial compartments necessary to support T-cell development. Interestingly, the involvement of epithelial cells and mesenchyme in both thymus organogenesis and repair suggests that common mechanisms operate during these processes. Moreover, the importance of thymic mesenchyme and its products, in this case, IL33, in thymus regeneration may reveal a potential cellular target for the therapeutic improvement of thymus function and immune reconstitution. Finally, and following on from observations that eosinophils are essential for thymus regeneration[10], our current study addresses the mechanism by which eosinophils aid in the recovery of thymus function, an important area that remained poorly understood. Importantly, we show here that IL4 administration can replace the need for eosinophils in eosinophil-deficient mice. This then strongly suggests that eosinophil production of this type 2 cytokine represents at least part of the effector mechanism that explains their involvement in thymus recovery. Indeed, such findings fit well with the importance of eosinophils and IL4 during the regulation of tissue recovery in other organs. These include liver[37] and nerve repair[38], and collectively point towards common mechanisms of regeneration following tissue injury in both lymphoid and non-lymphoid tissues.

## Methods

### Mice
Female mice at 8–10 weeks of age were used throughout this study. They were housed under barrier conditions in ventilated cage racks, and sacrificed by cervical dislocation. The animal facility is maintained under artificial lighting (12 h) between 7:00am and 7:00 pm, with a controlled ambient temperature of 22 °C ± 2 °C, and relative humidity between 45% and 65%. The following mice were on a Balb/c background: Balb/c WT, DdblGATA[28] (strain number 005653, Jackson Laboratories), Il4ra−/−[39], Il33cit/+ (IL33cit reporter) and Il33cit/cit (IL33KO)[17], IL25Tdtom/Tdtom (IL25 KO)[40], IL4GFP[24] (strain number 004190, Jackson Laboratories). The following mice were on a C57BL/6 background; Red5xRagGFP mice generated in-house by crossing IL5tdTomato reporter (Red5) (strain number 030926, Jackson Laboratories)[15] with Rag2pGFP transgenic[16] (strain number 005688, Jackson Laboratories), ILC2-deficient Il7raCreRorafl/fl mice were generated by crossing Il7raCre[41] and Rorafl/fl[42] mice. For experiments involving the comparison of thymus regeneration in WT, Il33−/− and Il25−/− mice, all animals were born and maintained in the Biomedical Services Unit at the University of Birmingham. For experiments analysing effects of IL25 and IL33 on

thymus regeneration in WT mice, Balb/c mice (strain code 028) were purchased from Charles River. Housing and experimental methods were performed at the Biomedical Services Unit at University of Birmingham, following approval by the local Animal Welfare and Ethical Review Body (AWERB) and UK National Home Office.

### Flow cytometry and cell sorting
For eosinophil analysis, cells were obtained from thymus tissue after enzymatic digestion using collagenase D (2.5 mg/ml) (Roche) and deoxyribonuclease I (DNase I) (100 mg/ml) (Roche) at 37 °C[10]. For thymocyte, NKT and ILC analysis, thymi were mechanically disaggregated before surface staining. Reagents used for eosinophils/iNKT/ILC were anti-CD45, clone 30-F11, Brilliant Violet 786 (1:100), APC-eFluor 780 (1:800), Brilliant 605 (1:800), eBioscience, anti-TCRβ clone H57-597, APC-eFluor 780 (1:200), eBioscience, anti-CD11b clone, M1/70, PE (1:400), BioLegend, anti–Siglec-F clone ES22-10D8, APC (1:200), Biotin (1:100) Miltenyi Biotech, anti-CD4 clone RM4-5, Brilliant Violet 711 (1:50), Alexa Fluor 700 (1:100), FITC (1:200), BioLegend, anti-CD8α clone 53-6.7, Brilliant Violet 510 (1:200), BioLegend, anti-TER119 clone TER-119, Brilliant Violet 421 (1:100), Alexa Fluor 700 (1:200) BioLegend), PBS57/mCD1d tetramer, APC (1:200), National Institutes of Health Tetramer Core Facility, anti-IL7Rα clone A7P34, Brilliant Violet 421 (1:100), APC (1:100), eBioscience, anti-KLRG1 clone 2F1/KLRG1, PerCP Cy5.5 (1:200), BioLegend and anti-NK1.1 clone PK136, Brilliant Violet 650 (1:100) Invitrogen. Lineage panel used for ILC analysis[10] was anti-CD3ε clone 145-2C11, PE Cy7 (1:200), Biolegend, anti-CD5 clone 53-7.3, PE Cy7 (1:200), eBioscience, anti-CD11b clone ICRF44, PE Cy7 (1:200), Invitrogen, anti-CD11c clone N418, PE Cy7 (1:200), Invitrogen, and anti-B220 clone RA3-6B2, PE Cy7 (1:200), Invitrogen. For TEC, mesenchyme, and endothelial analysis[10,43], thymus tissues were digested with collagenase dispase (2.5 mg/ml) (Roche) and DNase I (100 mg/ml) (Roche) before being stained with the following antibodies/reagents: anti-EpCAM1 clone G8.8, Brilliant Violet 711 (1:400), PerCP-eFluor 710 (1:2000), eBioscience, UEA1 Biotin (1:10000) (Vector Labs) detected using streptavidin PE Cy7 (1:1500), eBioscience, anti-Ly51 clone BP-1, PerCP-eFluor 710 (1:600), BD Pharmingen, anti-MHCII clone M5/114.15.2, Alexa Fluor 700 (1:1200), eBioscience, anti-CD80 clone 16-10A1, Brilliant Violet 605 (1:400), BioLegend, anti-CD31 clone 390, PE Cy7 (1:800), Invitrogen, anti-Sca-1 clone D7, PE (1:2000), FITC (1:1000), eBioscience, anti-IL4Ra clone I015F8, PE (1:50), BioLegend, anti-Integrinα7 clone 334908, Alexa Fluor 700 (1:100), R&D, and Zombie Aqua Fixable Viability Kit, Brilliant Violet 510 (1:1000), BioLegend. All flow cytometric data were collected on a BD LSR Fortessa running FACs DIVA (v.9.0) and analysed using FlowJo version v10.8.1.

### Confocal microscopy
Thymus sections of 7-μm thickness were cut from snap-frozen thymus tissue, then fixed in acetone. Sections were stained with the following antibodies: mTEC marker ERTR5[44], anti-Aire (5H12, Invitrogen), anti-CD205 (205yekta, Invitrogen). Secondary antibodies were streptavidin 555 (Invitrogen), goat anti-Rat IgM Alexa Fluor 647 (Invitrogen). Sections were counterstained with DAPI (4′,6-diamidino-2-phenylindole) (Sigma), and mounted using Prolong Diamond (Thermo Fisher). All confocal microscopy data was acquired on a Zeiss LSM 880 microscope using Zen Black (14.0.22.201) and analysed using Zen Black (16.0.2.306).

### Sub-lethal irradiation
Mice on a Balb/c background were subjected to sublethal total-body irradiation[10] with a dose of 1 × 425 rad (4.25 Gy). Mice on a C57BL/6 background were given 1 × 500 rad (5.0 Gy)[10] (MultiRad 350 x-ray irradiation system, RPS). Mice were given Baytril for 1 week before, and 1 week after total body SLI treatment. Mice were then harvested at indicated time points to assess thymus recovery.

## Recombinant cytokine injections

Mice were intraperitoneally injected with either PBS or 5 μg recombinant IL4 (PeproTech) complexed to 25 μg anti-IL4 (BioXCell) prior to injection, 2.5 μg of recombinant IL25 (PeproTech) or 2.5 μg of recombinant IL33 (PeproTech). In some experiments, mice received a combination of IL4 and IL33 described above. Injections and harvesting points are indicated in the figures.

## Statistical analysis

Graph Prism v 10.0.3 (GraphPad Software) was used to perform all statistical analyses. Unpaired Student's *t* test was used for comparisons between two data sets, and graphs were annotated to indicate significance. For comparison of three or more groups, one-way analysis of variance (ANOVA) was performed with an appropriate post hoc test specified in each figure legend to allow for multiple comparisons. Nonsignificant differences are not specified. In all figures, bar charts and error bars represent means ± SEM.

## Reporting summary

Further information on research design is available in the Nature Portfolio Reporting Summary linked to this article.

## Data availability

Authors confirm the data that supports the findings of this study are available in the figures and supplementary figures. Data generated in this study are provided in the accompanying Source Data file. Source data are provided with this paper.

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

## Acknowledgements

The authors thank BMSU staff for expert animal husbandry. We also thank Prof Hans Reimer Rodewald for *Il7ra*<sup>Cre</sup> mice, and the NIH Tetramer Facility for mCD1d-PBS57 tetramers. This work was supported by an MRC Programme Grant to GA (MR/T029765/1). A.N.J.M. is supported by the Medical Research Council, as part of United Kingdom Research and Innovation (UK Research and Innovation) (MRC grant U105178805).

## Author contributions

E.J.C. designed and performed experiments, analysed data, and wrote the manuscript. K.D.J., A.J.W., and S.M.P. designed and performed experiments; A.N.J.M. provided key reagents, expert advice and edited the manuscript; A.B. designed experiments and provided key reagents, G.A. and W.E.J. designed experiments, analysed data, and wrote the manuscript.

## Competing interests

The authors declare no competing interests.
