## [Peer Review File · Nature Communications]

REVIEWER COMMENTS

Reviewer #1 (expert in thymic epithelial cells):

In this paper, Cosway et al. examined the cellular and molecular mechanisms that control thymus regeneration after irradiation-induced atrophy, in particular focusing the role of thymic eosinophils and their effector type-2 cytokines. They claimed that the alarmin IL-33 produced by thymic mesenchymal cells is essential for the regeneration of damaged thymus, as monitored by the numbers of thymocytes and TECs. Their results show that administration of IL-33 in mice induced expansion of ILC2 cells that produce IL-5, which triggers the recruitment to the thymus and IL-4 production of eosinophils. The IL-4 receptor is highly expressed in mesenchymal cells in the thymus. IL-4 administration increased the numbers of thymic mesenchymal cells and promoted the thymus recovery following irradiation, bypassing the effector function of eosinophils. These data suggest the presence of a positive feedback mechanism of lympho-stromal interactions mediated by type-2 cytokines that control the thymus regeneration.

The mechanism proposed in this study is interesting, whereby lymphocytes and stromal cells respond to thymic damage in a coordinated and sequential manner and a group of type-2 cytokines link them together to promote thymic regeneration. However, there are some questionable aspects of their data, which are not merely technical insufficiency but may be important for the main message of this paper.

In particular, issues of cell numbers raised in the following comments, #1, #4 and #5, should be addressed, because the number of total thymus cells or TECs are used as major indicators of thymus regeneration in this study.

Major points:

#1

The authors show in Fig 2 that in vivo administered rIL33 increased the numbers of thymocytes and mTECs after SLI. However, the number of mTECs of control mice (injected with PBS) in Fig 2C is markedly lower than that of control mice in other Figures. For example, the PBS mice in Fig 2 have about 0.7×10^4 mTEChi cells, but the WT mice in Fig 1 and the PBS mice in Fig 3 have $1.5-1.6 \times 10^4$ mTEChi cells. The rIL33-treated mice in Fig 2 have approx. 1.5×10^4 mTEChi cells, which is similar to the number of mTEChi cells in WT mice in Fig 1 and PBS-treated or rIL25-treated mice in Fig 3. Thus, comparing across figures in this paper, it seems that only the PBS used in Fig 2 had a negative effect on mTEChi cells, which does not support the authors' interpretation that rIL33 promoted thymus regeneration.

#2

Did the i.p. injected cytokines (rIL-33, rIL-25 and rIL-4) reach the thymus? Do the authors have evidence that such cytokines affect directly lymphocytes or stromal cells in the thymus? Or, do they act somehow indirectly?

#3

In Fig 6D and E, the authors observed a marked and rapid decline in Sca1+ mesenchymal cells. However, it has long been known that thymic stromal cells including mesenchymal cells are resistant to irradiation. Indeed, although mTECs subsets exhibit reduction and recovery in response to ageing, pregnancy and stress, other types of thymic stromal cells do not show a severe reduction upon such stimuli. Is the authors' interpretation that Sca1+ mesenchymal cells decrease and recover prior to TECs convincing? The authors should present flow cytometry profiles analyzing the number of Sca1+ mesenchymal cells (as supplementary data).

#4

Fig 7B shows that, at day 1 after SLI, 14.7% of IL4GFP+ cells in the thymus are iNKT cells while 82.4% are eosinophils. These data are not consistent with those of the authors' previous paper (Cosway et al, Sci Immunol 2022; Fig 6C), where 87% of IL4GFP+ cells in the thymus are iNKT cells at day1 after SLI. The authors should address and explain the discrepancies in the data between their own studies.

#5

In Fig 7I; the number of total cells in PBS-treated delta-dblGATA mice is $\sim 0.5 \times 10^8$, but the number of thymocytes is less than 0.1×10^8 . The numbers of thymic stromal cells including TECs and mesenchymal cells are in the order of 10^4 to 10^5 . What are the remaining cells ($\sim 0.4 \times 10^8$ cells)? Clearly, the calculations do not match. It is possible that there are fatal errors in the flow cytometry data or calculation methods used to determine cell counts.

Minor points:

#6

Page 6, line 6:

'Intrathymic Expression Of IL33 Expression Is' -> 'Intrathymic Expression Of IL33 Is'

#7

In Fig 7I, the y-axis of the graph of total cells is incorrect.

#8

Page 13, line 1-2:

It is described that the authors used collagenase D (Roche) for preparation of thymic stromal cell suspension. Is it correct? As previous papers indicated, preparation of thymic stromal cells with collagenase D results in lower cell yield than with Liberase and possibly cause the loss of certain stromal cells (Seach et al, J Immunol Methods 2012; Nitta et al, Immunol Rev 2021). I recommend the authors not to use collagenase D for the preparation of thymic stromal cells, especially endothelial cells and pericytes, in future, unless there is a specific reason.

Reviewer #2 (expert in T cell and ILC2 development):

The manuscript by Cosway et al "The alarmin IL33 orchestrates Type 2 Immune-Mediated Control of Thymus Regeneration" describes the partial mechanism underlying the regeneration of the thymic epithelial compartment and thymic function after irradiation. The authors previously identified thymic eosinophils as essential players in this regeneration. It is now shown that IL33, produced by Sca-1 expressing stromal cells, expands thymic ILC2 that are essential for regeneration. In the absence of ILC2, IL33 alone is not capable to restore neither thymic function nor eosinophilia. However, IL4 alone can restore mesenchymal cell numbers and IL33 production, after irradiation and in steady-state even in the apparent absence of eosinophils. Curiously, NKT cells that are the main producers of IL4 in the thymus are overridden by eosinophils in IL4 production, after irradiation.

It has been shown in a previous publication from this group that IL5 alone can drive epithelial cell regeneration in the absence of ILC2 or eosinophils. From that publication (Cosway et al 2022 Science Immunol.) one gets the idea that IL5 made by eosinophils or ILC2, mediate regeneration. NKT cells producing IL4 would trigger CCL11 by epithelial cells that would attract eosinophils that together with ILC2 directly induce regeneration through IL5. However, in addition to IL5, ILC2 also produce IL4 and IL13. Could these be involved in the process?

Main point

It is somewhat difficult to integrate the results from both publications and an effort should be made to

reconcile them. What is finally the role of eosinophils? Because IL4 can now restore mesenchymal cell numbers and presumably also IL33 that stimulates ILC2 leading to restored epithelia and all of this in the absence of eosinophils, maybe not much. NKT are radiation resistant and main producers of IL4. Also, in the previous publication NKT D1 after irradiation are still the main producers of IL4 whereas in this manuscript they are not, this is confusing. How many ILC2 are found in the thymus after irradiation and how much IL5 do they produce?

Concerns.

1. Since the authors have the IL4 reporter it would be interesting to compare the IL4 expression before and after injury in the three populations (NKT, eosinophils and ILC2). Numbers of ILC2 in the thymus in the same time points should be shown.
2. The authors show IL4 GFP in NKT and eosinophils after injury. What about the non IL4 expressing NKT and eosinophils. In a previous publication (Science Immunology Fig 6) it is shown that at D0 and D1 after SLI the main IL4 producers are the NKT and not the eosinophils, as shown here. This point needs clarification.
3. Although *dbGATA* mice are devoid of eosinophils due to a deletion in the palindromic DNA binding site in the GATA1 promoter, bone marrow progenitors can be induced by cytokine stimulation *in vitro* to produce eosinophils (Dyer et al J. Immunol. 2007). It would be important to show that no eosinophils develop in mutant mice after treatment with recombinant IL4.

Reviewer #3 (expert in thymus biology and function):

Exploring the alarmin IL33's role on the recovery of sublethal irradiation (SLI)-induced acute thymic injury, Cosway et al. reveal a positive feedback mechanism of type-2 innate immunity, in which Sca1+ thymic mesenchyme-produced IL33 induces ILC2 cell expansion to trigger thymic eosinophil to produce IL4 for re-establishing thymic mesenchyme, thereafter recovering thymic epithelial compartment. This is an elegant work with excellent writing. However, if the authors can address the following several concerns, the paper should be more impressive.

1. This finding is in an acute thymic injury and regeneration via a sublethal irradiation (SLI) model, but not age-related chronic thymic injury. This point should be clearly mentioned, since the authors did not have any evidence that IL33 or IL4 can rejuvenate age-related thymic injury. Both acute and chronic thymic injuries have different mechanisms and pathology. If the authors can test a natural aged model with IL33, that will be great. However, if not, the authors should clearly state that this positive feedback mechanism of type-2 innate immunity may not apply for the age-related chronic thymic injury.
2. In most results, the authors presented changes in thymic cell numbers (hematopoietic and non-hematopoietic lineages) with flow cytometry. This is very good, but it is insufficient and superficial. Particularly for the non-hematopoietic lineage cells, after enzymatic dissociation/isolation, FACS cell-surface stained TECs (without using any intrinsic TEC-lineage markers, such as *Foxn1-GFP*) usually lose their bona fide features, including true cell numbers. Immunofluorescence staining of thymic sections by analysis with confocal-microscopy for non-hematopoietic lineage thymic cells in the thymic microstructure will provide more bona fide information related to the injury and recovery.
3. This work involves various and multiple genetic knockout mouse models. This enhances data persuasion. However, why in some experiments Balb/c mice were used, while in others B6 mice were used. Please justify whether IL33 rejuvenation effects at different genetic background mice are the same?
4. Based on the positive feedback pathway proposed by the authors, IL33 acts upstream of the pathway and IL4 works downstream of the pathway. Which cytokine is better for recovery from SLI-induced acute thymic injury? Whether will use both of them in a cocktail style achieve the best recovery effects?

RE: NCOMMS-23-23959

The Alarmin IL33 Orchestrates Type 2 Immune-Mediated Control Of Thymus Regeneration

Dear ,

Thank you for your email dated 3rd July and providing us with reviews of our manuscript. We provide a point-by-point response below, where comments are addressed and discussed fully. Individual reviewer's comments are highlighted in bold, and our specific responses are below. Revisions in the manuscript are highlighted in yellow.

Reviewer 1.

We thank the reviewer for highlighting our study as interesting, and for summarizing our findings on a new mechanism for thymus regeneration.

- 1. The authors show in Fig 2 that in vivo administered rIL33 increased the numbers of thymocytes and mTECs after SLI. However, the number of mTECs of control mice (injected with PBS) in Fig 2C is markedly lower than that of control mice in other Figures. For example, the PBS mice in Fig 2 have about 0.7×10^4 mTEChi cells, but the WT mice in Fig 1 and the PBS mice in Fig 3 have $1.5-1.6 \times 10^4$ mTEChi cells. The rIL33-treated mice in Fig 2 have approx. 1.5×10^4 mTEChi cells, which is similar to the number of mTEChi cells in WT mice in Fig 1 and PBS-treated or rIL25-treated mice in Fig 3. Thus, comparing across figures in this paper, it seems that only the PBS used in Fig 2 had a negative effect on mTEChi cells, which does not support the authors' interpretation that rIL33 promoted thymus regeneration.**

This point relates to comparison of data shown in Figures 1, 2 and 3, and our conclusion that rIL33 promotes thymus regeneration. We feel there are two reasons for differences in cell numbers pointed out by the reviewer. The first relates to experimental design, and we apologise for not making this clear in our initial submission. In Figure 1, we compare thymus regeneration in Il25^{-/-} mice and Il33^{-/-} mice with WT controls. Because Il25^{-/-} and Il33^{-/-} mice were born and maintained in our local animal facility, WT controls in these

experiments were also born and maintained in our facility. This allowed for direct comparison between WT and KO mice sourced and housed in the same way. For Figures 2 and 3, we used commercially available WT mice that were injected with IL25, IL33 or PBS. This was necessary to consistently obtain suitable group sizes of mice to perform these experiments, and ensured mice used for experiments in Figures 2 and 3 were housed in the same way. Thus, control and experimental mice in Figure 1 were suitably matched, and control and experimental mice in Figures 2 and 3 were also suitably matched. Because of this difference in experimental design, comparison of data in Figure 1 with that in Figures 2 and 3 is not possible. Again, we apologise for not clarifying this, which is now explained on pages 4 and 5 and in the Materials and Methods section. Regarding the second reason, we agree with the reviewer that data shown in Figure 2 is important as it relates to our conclusion that IL33 boosts thymus regeneration. On close scrutiny of data 35d post-SLI, we agree that cell numbers in Figure 2C are lower than that in Figure 3C (e.g., Figure 2C total thymus cells following PBS injection: 1×10^8 ; Figure 3C total thymus cells following PBS injection 1.5×10^8), which likely impacts enumeration of individual cell types in these experiments. Thus, even though mice used for Figures 2 and 3 were from the same commercial source, a reason for this numerical difference may be that animals used for IL33 administration and d35 analysis (Figure 2C) were suboptimal. Because of this, we have now repeated experiments involving IL33 administration in new batches of commercially available WT Balb/c mice at d35 post-SLI. Importantly, these new data align well with similar control experiments in Figure 3C and support our conclusion that IL33 administration boosts thymus regeneration. This new data shown in Figure 2C has now replaced our initial data.

2. Did the i.p. injected cytokines (rIL-33, rIL-25, and rIL-4) reach the thymus? Do the authors have evidence that such cytokines affect directly lymphocytes or stromal cells in the thymus? Or, do they act somehow indirectly?

Using an i.p. route to inject mice with cytokines is a commonly used approach to study mechanisms of thymus regeneration. However, in our study and the studies of others, we agree that it is unclear whether these reagents exert their effects solely intrathymically, or via additional indirect effects in the periphery. Relevant to this, we show in this study that cellular targets of IL33 (ILC2) and IL4 (thymic mesenchyme) are present intrathymically, and previously showed i.p. IL25 increased thymic IL25R⁺ NKT-cells (Nat Comms 2020 11:2198), suggesting i.p. cytokine administration is likely to include direct action on thymus. However, we also include discussion of our findings to accommodate the possibility of indirect extrathymic effects, on page 12.

3. In Fig 6D and E, the authors observed a marked and rapid decline in Sca1+ mesenchymal cells. However, it has long been known that thymic stromal cells including mesenchymal cells are resistant to irradiation. Indeed, although mTECs subsets exhibit reduction and recovery in response to ageing, pregnancy and stress, other types of thymic stromal cells do not show a severe reduction upon such stimuli. Is the authors' interpretation that Sca1+ mesenchymal cells decrease and recover prior to TECs convincing? The authors should present flow cytometry profiles analyzing the number of Sca1+ mesenchymal cells (as supplementary data).

There are several reports that showing that in addition to haemopoietic cells, thymic stromal cells are also sensitive to irradiation (Adkins et al J. Immunol. 1988, 140:3373; Chung et al Blood 2001, 98:1601; Kelly et al Blood 2010 115: 1088). While other studies may have not specifically analysed thymic mesenchyme, we feel our study extends these earlier observations by showing that mesenchyme is indeed a target of irradiation, which is

consistent with these earlier findings indicating sensitivity within the thymic stromal compartment. To re-inforce the observation that mesenchymal cells are damaged post-SLI, we also now include numeration of total thymic mesenchyme following irradiation and during thymus regeneration (Supp. Figure 4). Moreover, we have now added new statistical analysis to analysis of the loss and recovery of Sca1⁺ mesenchyme, which we feel further emphasizes the robustness of our conclusion (Figure 6D). As requested, we also now show flow cytometry profiles of Sca1⁺ mesenchyme cells used to quantitate their number post-SLI (Supp. Figure 4). This is discussed on page 7.

- 4. Fig 7B shows that, at day 1 after SLI, 14.7% of IL4GFP⁺ cells in the thymus are iNKT cells while 82.4% are eosinophils. These data are not consistent with those of the authors' previous paper (Cosway et al, Sci Immunol 2022; Fig 6C), where 87% of IL4GFP⁺ cells in the thymus are iNKT cells at day1 after SLI. The authors should address and explain the discrepancies in the data between their own studies.**

This point relates to how our current findings relate to those in our earlier paper (Science Immunology 2022). In Figure 6 of our 2022 Science Immunology paper, we showed that at d0 and d1 most IL4^{GFP+} cells are NKT-cells. In our current manuscript, Figure 7A shows that at d0 most IL4^{GFP+} cells are NKT, while at d1 most IL4^{GFP+} cells are eosinophils. The reasons for these differences likely relate to the background strain of the IL4^{GFP} mice used in each study. As stated in the Materials and Methods section of the Science Immunology paper, we used IL4^{GFP} mice on a B6 background, as this was the only version available at the time. In our current manuscript, and as stated in the Materials and Methods section, we used IL4^{GFP} mice on a Balb/c background. As IL4^{GFP} Balb/c mice became available to us during our most recent work, we felt it was important to use them so that the Balb/c background was consistent between IL4 reporters, Δ dblGATA eosinophil deficient mice and *Il4ra*^{-/-} mice that are key strains in this study. We believe these differing observations may indicate that there are some differences in the thymus regeneration mechanisms in Balb/c and B6 mice, which would fit well with the known differences in type 1 and type 2 immunity in these mice, particularly in relation to the intrathymic source of IL4 and frequency of NKT-cells. We now discuss this on pages 10-11. Importantly, we feel this does not detract from the findings in our new study, where we are able for the first time to compare IL4 availability/eosinophil requirement/IL4R α requirement for thymus regeneration in Balb/c mice, and show that i) Balb/c mice lacking eosinophils have failures in thymus regeneration (our previous study, and shown again here), ii) in Balb/c mice eosinophils are the major producers of IL4 post-SLI (this manuscript), and iii) that IL4 can rescue thymus regeneration in eosinophil deficient mice on a Balb/c background (this manuscript). In the discussion, we emphasise that our current work is predominantly focused on thymus regeneration in Balb/c mice and highlight potential differences between thymus regeneration in Balb/c and B6 mice to reflect findings of both of our studies.

- 5. In Fig 7I; the number of total cells in PBS-treated delta-dblGATA mice is $\sim 0.5 \times 10^8$, but the number of thymocytes is less than 0.1×10^8 . The numbers of thymic stromal cells including TECs and mesenchymal cells are in the order of 10^4 to 10^5 . What are the remaining cells ($\sim 0.4 \times 10^8$ cells) ? Clearly, the calculations do not match. It is possible that there are fatal errors in the flow cytometry data or calculation methods used to determine cell counts.**

We apologise for this; the explanation is numerical miscalculation due to human error. We have now corrected this in the revised Figure 7K.

6. Page 6, line 6: ‘Intrathymic Expression Of IL33 Expression Is’ -> ‘Intrathymic Expression Of IL33 Is’

This has now been corrected.

7. In Fig 7I, the y-axis of the graph of total cells is incorrect.

This has now been corrected (now Figure 7K).

8. Page 13, line 1-2: It is described that the authors used collagenase D (Roche) for preparation of thymic stromal cell suspension. Is it correct? As previous papers indicated, preparation of thymic stromal cells with collagenase D results in lower cell yield than with Liberase and possibly cause the loss of certain stromal cells (Seach et al, J Immunol Methods 2012; Nitta et al, Immunol Rev 2021). I recommend the authors not to use collagenase D for the preparation of thymic stromal cells, especially endothelial cells and pericytes, in future, unless there is a specific reason.

We respectfully point out that for preparation of thymic stromal cells, we use collagenase dispase, and not collagenase D as stated by the reviewer. This is stated in the methods section and is also the method used in our previous study on eosinophils and thymus regeneration to ensure consistency (Cosway et al Science Immunol. 2022).

Reviewer 2.

We thank the reviewer for emphasizing how our new study expands on our earlier work to identify the importance of IL33 in thymus regeneration.

1. It has been shown in a previous publication from this group that IL5 alone can drive epithelial cell regeneration in the absence of ILC2 or eosinophils. From that publication (Cosway et al 2022 Science Immunol.) one gets the idea that IL5 made by eosinophils or ILC2, mediate regeneration. NKT cells producing IL4 would trigger CCL11 by epithelial cells that would attract eosinophils that together with ILC2 directly induce regeneration through IL5. However, in addition to IL5, ILC2 also produce IL4 and IL13. Could these be involved in the process?

The reviewer raises the interesting possibility that ILC2 production of type 2 cytokines might also be involved in thymus regeneration. We have now looked for IL4 expression by thymic ILC2, and see that in the steady state, approximately 60% of ILC2 express GFP in IL4^{GFP} reporter mice (this new data is shown in Figure 7E). Importantly, in answer to point 2 by this reviewer (see below), we also analysed ILC2 frequency after damage, and found that ILC2 are absent from the thymus 1 day after irradiation. Collectively these new data indicate that despite their presence and IL4 production in steady state, ILC2 involvement is limited to the first 24 hours after damage, and after this point would be unlikely to act as a source of IL4. These new data are included in Figure 7E, F, and S. Figure 6, and described on pages 9 and 11.

2. It is somewhat difficult to integrate the results from both publications and an effort should be made to reconcile them. What is finally the role of eosinophils? Because IL4 can now restore mesenchymal cell numbers and presumably also IL33 that stimulates ILC2 leading to restored epithelia and all of this in the absence of eosinophils, maybe not much. NKT are radiation resistant and main producers of IL4. Also, in the previous publication NKT D1 after irradiation are still the main producers of IL4 whereas in this manuscript they are not, this is confusing. How many ILC2 are found in the thymus after irradiation and how much IL5 do they produce?

Several points are made here.

- i) On the comparison of our new manuscript to our initial report (Sci Immunol. 2022): this relates to how our current findings compare to those in our earlier paper (Science Immunology 2022). In our earlier work, Figure 6 showed that at d0 and d1 most IL4^{GFP+} cells are NKT-cells, while in our new work Figure 7A shows that at d0 most IL4^{GFP+} cells are NKT, while at d1 most IL4^{GFP+} cells are eosinophils. The reasons for these differences likely relate to the background strain of the IL4^{GFP} mice used in each study. As stated in the Materials and Methods section of the Science Immunology paper, we used IL4^{GFP} mice on a B6 background, as this was the only version available at the time. In our current manuscript, and as stated in the Materials and Methods section, we used IL4^{GFP} mice on a Balb/c background. As IL4^{GFP} Balb/c mice became available to us during our most recent work, we felt it was important to use them so that the Balb/c background was consistent between IL4 reporters, Δ dblGATA eosinophil deficient mice and *Il4ra*^{-/-} mice that are key strains in this study. We believe these differing observations may indicate that there are some differences in the thymus regeneration mechanisms in Balb/c and B6 mice, which would fit well with the known differences in type 1 and type 2 immunity in these mice, particularly in relation to the intrathymic source of IL4 and frequency of NKT-cells. We now discuss this on page 10-11. Importantly, we feel this does not detract from the findings in our new study, where we are able for the first time to compare IL4 availability/eosinophil requirement/IL4R α requirement for thymus regeneration in Balb/c mice, and show that i) Balb/c mice lacking eosinophils have failures in thymus regeneration (our previous study, and shown again here), ii) in Balb/c mice eosinophils are the major producers of IL4 post-SLI (this manuscript), and iii) that IL4 can rescue thymus regeneration in eosinophil deficient mice on a Balb/c background (this manuscript). In the discussion, we emphasise that our current work is predominantly focused on thymus regeneration in Balb/c mice and highlight potential differences between thymus regeneration in Balb/c and B6 mice to reflect findings of both of our studies.
- ii) Regarding the role of eosinophils, it is also important to note that our previous study did not elucidate the role of eosinophils in thymus regeneration, so this remained a key unanswered question. Here, we show here that thymus regeneration in eosinophil-deficient mice can be rescued by IL4. This represents a novel finding that explains the requirement for eosinophils in thymus regeneration through their IL4 production. Importantly, this parallels how eosinophils regulate regeneration of other organs suggesting a common mechanism for recovery from injury in multiple tissues. We further emphasise these points, and cite relevant literature, on page 13.
- iii) Regarding the final sentence (See also point 1 of this reviewer), as requested we have also examined the frequency of thymic ILC2 after irradiation. In contrast to steady-state, we find that ILC2 are absent from the thymus 1 day after irradiation, and so IL5-expressing ILC2 were undetectable. This is of interest as it suggests ILC2 exert an important effect on thymus regeneration immediately (i.e., within the first 24 hours) after damage. This new data is now shown in Figure 7E, F and Supp. Fig. 6 and described on pages 9 and 11.

Concerns.

1. Since the authors have the IL4 reporter it would be interesting to compare the IL4 expression before and after injury in the three populations (NKT, eosinophils and ILC2). Numbers of ILC2 in the thymus in the same time points should be shown.

This point relates to point 1 by this reviewer, on analysis of IL4 expression by ILC2 in both steady state and after irradiation, as well as numbers of ILC2 in steady-state and post-SLI.

As stated previously, in new experiments we find that some ILC2 express IL4^{GFP} in the steady-state, and that ILC2 are depleted from the thymus 1-day post-SLI. This is now included alongside analysis of eosinophils and NKT-cells as requested in Figure 7A-F and S. Figure 6 and described on pages 9 and 11.

2. The authors show IL4 GFP in NKT and eosinophils after injury. What about the non IL4 expressing NKT and eosinophils. In a previous publication (Science Immunology Fig 6) it is shown that at D0 and D1 after SLI the main IL4 producers are the NKT and not the eosinophils, as shown here. This point needs clarification.

This point relates to findings in this manuscript and those in our previous manuscript, on the frequencies of NKT and eosinophils at D0 and D1 post SLI. We refer to our response to the same issue (point 2) above.

3. Although dbIGATA mice are devoid of eosinophils due to a deletion in the palindromic DNA binding site in the GATA1 promoter, bone marrow progenitors can be induced by cytokine stimulation in vitro to produce eosinophils (Dyer et al J. Immunol. 2007). It would be important to show that no eosinophils develop in mutant mice after treatment with recombinant IL4.

This is an important point, and as suggested we have now performed new experiments which show that IL4 treatment does not rescue eosinophil development in Δ dbIGATA mice (Supp. Figure 7) and discussed on pages 9-10.

Reviewer 3.

We thank this reviewer for describing our study as '*elegant work with excellent writing*'. As suggested, we have carefully followed the reviewer's advice to make our paper even more impressive.

1. This finding is in an acute thymic injury and regeneration via a sublethal irradiation (SLI) model, but not age-related chronic thymic injury. This point should be clearly mentioned, since the authors did not have any evidence that IL33 or IL4 can rejuvenate age-related thymic injury. Both acute and chronic thymic injuries have different mechanisms and pathology. If the authors can test a natural aged model with IL33, that will be great. However, if not, the authors should clearly state that this positive feedback mechanism of type-2 innate immunity may not apply for the age-related chronic thymic injury.

We agree that thymus injury can be both acute and chronic and understand that studying both is important. We also acknowledge that our study focusses on mechanisms that regulate regeneration following acute injury. As requested, we emphasis this focus of our findings, and cite literature relevant to thymus ageing and chronic injury, on page 12.

2. In most results, the authors presented changes in thymic cell numbers (hematopoietic and non- hematopoietic lineages) with flow cytometry. This is very good, but it is insufficient and superficial. Particularly for the non-hematopoietic lineage cells, after enzymatic dissociation/isolation, FACS cell-surface stained TECs (without using any intrinsic TEC-lineage markers, such as Foxn1-GFP) usually lose their bona fide features, including true cell numbers. Immunofluorescence staining of thymic sections by analysis with confocal microscopy for non-hematopoietic lineage thymic cells in the thymic microstructure will provide more bona fide information related to the injury and recovery.

We agree that analysis of 'non-haemopoietic lineage thymic cells' in intrathymic

microenvironments via confocal microscopy is important to understand the process of thymus regeneration. As suggested, we have now performed new experiments to look at thymic microenvironments via microscopy, by analysing thymic tissue sections of mice undergoing thymus regeneration following IL33 administration. Importantly, these experiments show that cTEC, mTEC and Aire⁺ mTEC are distributed appropriately within cortex and medulla areas, demonstrating the ability of IL33 to boost thymus regeneration occurs alongside the retention of normal thymus organization. These findings are described on page 5 and shown in Supp. Fig. 1.

3. This work involves various and multiple genetic knockout mouse models. This enhances data persuasion. However, why in some experiments Balb/c mice were used, while in others B6 mice were used. Please justify whether IL33 rejuvenation effects at different genetic background mice are the same?

We acknowledge the importance of maintaining consistency of mouse background and highlight that apart from ILC2 KO mice and Red5Rag2GFP mice, the other 5 strains of genetically manipulated mice are on the Balb/c background. The need to use some mice on a B6 background stems from practical availability of mice from our collaborators. As requested, we have now performed new experiments to compare the effects of IL33 on thymus regeneration of mice of different genetic backgrounds. Importantly, these new experiments show that IL33 boosts thymus regeneration in both B6 and Balb/c mice. This new data is shown in Supp. Fig. 2 and explained on page 5.

4. Based on the positive feedback pathway proposed by the authors, IL33 acts upstream of the pathway and IL4 works downstream of the pathway. Which cytokine is better for recovery from SLI-induced acute thymic injury? Whether will use both of them in a cocktail style achieve the best recovery effects?

We agree with the reviewer's interpretation of our findings and thank them for the interesting ideas of comparing effects of IL4 and IL33 on thymus regeneration and assessing potential synergistic effects. We have now performed new experiments where effects of treatment with IL4 alone, IL33 alone, or IL4+IL33 in combination were examined. Interestingly, we find that while IL4 and IL33 boost thymus regeneration to a similar extent, a combination of IL4+IL33 did not further enhance regeneration. This new data is shown in Supp. Fig. 8 and described on pages 11-12.

In conclusion, we thank the reviewers for their comments that have allowed us to include new data and further discussion/interpretation in our manuscript. This has further strengthened our study and helped us provide further insight into important mechanisms of thymus regeneration. We look forward to your consideration of our revised manuscript.

Yours Sincerely,

REVIEWERS' COMMENTS

Reviewer #1 (expert in thymic epithelial cells):

#1

For Figure 2, the authors acknowledged that the previous data were inadequate and provided the new data of IL-33 administration experiments using new 'batches' of purchased WT BALB/c mice. Surprisingly, in the present data, the number of TECs increased threefold in both the PBS and IL-33 groups compared to the previous data. It is usually unlikely to obtain such different results for different batches of commercially available mice. Very unnatural and unreasonable. Scrutiny of the data using the same WT BALB/c mice may also be necessary. Even in the newly provided Figure S1, there does not appear to be an increase in TECs. Is the authors' interpretation that IL-33 administration boosts thymus regeneration appropriate?

#2

The previous study the authors cited (Nat Commun 2020) does not demonstrate that i.p. injected cytokines reach the thymus. It showed that i.p. IL-25 resulted in an increase of thymic NKT2 cells and that IL-25 did not influence NKT cells in vitro, suggesting that i.p. administered IL-25 may even have affected thymic NKT2 cells via some extra-thymic events. The authors should not only discuss this issue but also provide data that the administered cytokines reach the thymic microenvironment. For example, in the experiments in Figures 2 and 3, can you detect increased levels of IL-33 or IL-25 in the tissue fluid from the thymus of administered mice by ELISA?

#3

The authors cite three papers to show that thymic mesenchymal cells are sensitive to irradiation, but all of those papers described thymic epithelial cells (especially mTECs) and do not support the authors' findings that Sca1+ thymic mesenchymal cells decrease after irradiation. The authors provided flow cytometry data in Figure S4 upon my request. However, the gating in the EpCAM-1 vs CD45 plot differs between on day 1 and others. The bottom panel shows the frequency of Sca1+ mesenchymal cells among TER119- CD45- EpCAM1- CD31- Itga7- gated cells, but the expression profiles of TER119, CD31 and Itga7 are not shown. In Figure 6D and E, Sca1+ mesenchymal cells are reduced to less than 20% on day 1 compared to day 0, but such trend cannot be obtained from the data presented by the authors in Figure S4. These data are not convincing.

#4

As other reviewers pointed out, difference in the genetic background of the mice (B6 or BALB/c) have resulted in confusion for readers (reviewers) and reduced reliability for the authors. The authors explained that the genetic background of the IL4GFP mice used in this study is BALB/c, so unlike the results from the previous study (Science Immunology 2022) using IL4GFP mice with B6 background, the main producer of IL-4 after SLI is eosinophils rather than NKT cells. However, Figure 1C of the Sci Immunol 2022 paper shows a 10-fold increase in thymic eosinophils in WT BALB/c mice on day 1 after SLI. Do these eosinophils not produce IL-4?

Furthermore, in the Sci Immunol 2022 paper, the authors showed that NKT-deficient mice on both BALB/c and B6 background exhibited significantly reduced thymus regeneration after SLI, arguing against a strain-specific action of thymus regeneration. These authors' own previous results contradict the statement in the present paper that SLI irradiation and thymus regeneration in BALB/c mice results in a shift in the cellular source in IL-4 production towards eosinophils.

#5

The authors properly addressed my criticism.

#8

This was my misreading. I realize that the authors used collagenase/dispase for preparation of thymic stromal cells.

Reviewer #2 (expert in T cell and ILC2 development):

The authors addressed all points raised by this reviewer. Specifically, the authors did a good job in integrating previous results and clarifying inconsistencies.

Reviewer #3 (expert in thymus biology and function):

The authors addressed all my concerns. I believe it meets the standards of Nature Communications and suggest to publish this manuscript in Nature Communications.

RE: NCOMMS-23-23959A

The Alarmin IL33 Orchestrates Type 2 Immune-Mediated Control Of Thymus Regeneration

Dear,

Thank you for providing us with reviews of our manuscript, and we are delighted to hear that it has now been accepted in principle. As requested, we provide a point-by-point response to reviewers comments below. Individual reviewer's comments are highlighted in track changes.

Reviewer 1.

#1 For Figure 2, the authors acknowledged that the previous data were inadequate and provided the new data of IL-33 administration experiments using new 'batches' of purchased WT BALB/c mice. Surprisingly, in the present data, the number of TECs increased threefold in both the PBS and IL-33 groups compared to the previous data. It is usually unlikely to obtain such different results for different batches of commercially available mice. Very unnatural and unreasonable. Scrutiny of the data using the same WT BALB/c mice may also be necessary. Even in the newly provided Figure S1, there does not appear to be an increase in TECs. Is the authors' interpretation that IL-33 administration boosts thymus regeneration appropriate?

This is a rehash of a comment from round 1 review. As suggested in the initial review, we performed new experiments to look at effects of IL33 on thymus regeneration. We point out that data shown in Fig 2 and S Fig2 clearly shows statistically significant increase in TECs following IL33 treatment. When mentioning Figure S1, the reviewer presumably means Figure S2? Again, we point out that both Fig 2 and Fig S1 show enhanced thymus regeneration after IL33 and SLI treatment. On the basis of the data presented, that involves both gain of function (ie IL33 injection) and loss of function (IL33 KO mice), we fully believe our data interpretation that IL33 is an important regulator of thymus regeneration. This approach is in line with the opinion of the editorial office, who state they 'did not find the new data unreasonable'. This is further explained on page 5.

#2 The previous study the authors cited (Nat Commun 2020) does not demonstrate that i.p. injected cytokines reach the thymus. It showed that i.p. IL-25 resulted in an increase of thymic NKT2 cells and that IL-25 did not influence NKT cells in vitro, suggesting that i.p. administrated IL-25 may even have affected thymic NKT2 cells via some extra-thymic events. The authors should not only discuss this issue but also provide data that the administrated cytokines reach the thymic microenvironment. For example, in the experiments in Figures 2 and 3, can you detect increased levels of IL-33 or IL-25 in the tissue fluid from the thymus of administrated mice by ELISA?

Again, we highlighted in our initial response that i.p. injected cytokines could have effects on the thymus via intrathymic and extrathymic mechanisms. We refer to this text on page 12, where we further highlight possible extrathymic effects.

#3 The authors cite three papers to show that thymic mesenchymal cells are sensitive to irradiation, but all of those papers described thymic epithelial cells (especially mTECs) and do not support the authors' findings that Sca1+ thymic mesenchymal cells decrease after irradiation. The authors provided flow cytometry data in Figure S4 upon my request. However, the gating in the EpCAM-1 vs CD45 plot differs between on day 1 and others. The bottom panel shows the frequency of Sca1+ mesenchymal cells among TER119- CD45- EpCAM1- CD31- Itga7- gated cells, but the expression profiles of TER119, CD31 and Itga7 are not shown. In Figure 6D and E, Sca1+ mesenchymal cells are reduced to less than 20% on day 1 compared to day 0, but such trend cannot be obtained from the data presented by the authors in Figure S4. These data are not convincing.

The reason we cited these 3 papers is to indicate the current status of the literature on the impact of irradiation on thymic stromal cells, which predominantly relates to TECs. We believe our study, demonstrating the sensitivity of mesenchyme to irradiation, adds to and extends current knowledge in this area. We now include full pre-gating of mesenchyme as requested. Furthermore, and in agreement with the editorial office, who state they 'could not identify any issues with Fig. 6D and S4.' we do not think that data shown in Fig 6D and S4 are incompatible with one another, as both show a convincing and statistically robust decline and then recovery in thymic mesenchyme caused by irradiation. To further clarify this, we now include discussion on p13 that highlights previous uncertainty of the effects of radiation on thymic mesenchyme.

#4 As other reviewers pointed out, difference in the genetic background of the mice (B6 or BALB/c) have resulted in confusion for readers (reviewers) and reduced reliability for the authors. The authors explained that the genetic background of the IL4GFP mice used in this study is BALB/c, so unlike the results from the previous study (Science Immunology 2022) using IL4GFP mice with B6 background, the main producer of IL-4 after SLI is eosinophils rather than NKT cells. However, Figure 1C of the Sci Immunol 2022 paper shows a 10-fold increase in thymic eosinophils in WT BALB/c mice on day 1 after SLI. Do these eosinophils not produce IL-4? Furthermore, in the Sci Immunol 2022 paper, the authors showed that NKT-deficient mice on both BALB/c and B6 background exhibited significantly reduced thymus regeneration after SLI, arguing against a strain-specific action of thymus regeneration. These authors' own previous results contradict the statement in the present paper that SLI irradiation and thymus regeneration in BALB/c mice results in a shift in the cellular source in IL-4 production towards eosinophils.

substantially, I think it would be preferable to maintain consistency).

This comment represents repetition of the reviewers initial review, and we addressed this point in our revised manuscript by extensively highlighting the importance of strain differences between B6 and BALB/c mice in the context of thymus regeneration. This is then directly relevant to the data and conclusions in our earlier paper (Science Immunology 2022) that used predominantly BALB/c

mice, and this current manuscript that uses predominantly B6 mice. We have further highlighted these points on page 11.

#5 The authors properly addressed my criticism.

No response required.

#8 This was my misreading. I realize that the authors used collagenase/dispase for preparation of thymic stromal cells.

No response required.

Reviewer #2

The authors addressed all points raised by this reviewer. Specifically, the authors did a good job in integrating previous results and clarifying inconsistencies.

We thank the reviewer for their support.

Reviewer #3

The authors addressed all my concerns. I believe it meets the standards of Nature Communications and suggest to publish this manuscript in Nature Communications.

We thank the reviewer for their support.

Yours Sincerely,